# Morphological Changes in Betulin Particles as a Result of Polymorphic Transformations, and Formation of Co-Crystals under Heating

**Svetlana A. Myz [1], Anatoly A. Politov [1], Svetlana A. Kuznetsova [2]** and **Tatyana P. Shakhtshneider [1,\*]**

1   Institute of Solid State Chemistry and Mechanochemistry SB RAS, Kutateladze Str., 18, 630090 Novosibirsk, Russia; apenina@solid.nsc.ru (S.A.M.); anpolitov@yahoo.com (A.A.P.)
2   Institute of Chemistry and Chemical Technology, FRC KSC SB RAS, Akademgorodok, 50/24, 660036 Krasnoyarsk, Russia; kuznetssvetl@yandex.ru
*   Correspondence: shah@solid.nsc.ru; Tel.: +7-383-332-53-44

**Abstract:** Changes in the morphology of betulin crystals during heating at selected temperatures corresponding to polymorphic transformations were investigated. It was shown that the prismatic crystals of starting betulin form III were converted into needles at 120 °C after water removal, followed by the III→II polymorphic transformation. During further heating up to 180 °C, the whiskers of betulin form I were grown. Experiments on betulin heating in the presence of dicarboxylic acids, adipic or suberic showed that the morphological changes can serve as a test for the formation of cocrystals. According to morphological changes, the formation of cocrystals of betulin with adipic acid under heating was identified. The interaction of adipic acid vapor with the surface of betulin crystals was suggested. In contrast, morphological changes in the mixture of betulin and suberic acid under heating provided only the evidence of polymorphic transformations of the components. The results on cocrystal formation by heating were compared with the preparation of cocrystals by the liquid-assistance grinding method. Despite the fact that polymorphic forms with a high surface area were formed when betulin was heated, dissolution studies showed that the starting betulin polymorph III exhibited the highest dissolution rate in comparison with betulin polymorphs obtained under heating.

**Keywords:** betulin; cocrystals; polymorphic transformation

## 1. Introduction

Betulin, lup-20(29)-ene-3β,28-diol, extracted from birch bark, exhibits various types of pharmacological activity, including antiviral, antibacterial, anti-inflammatory, anti-HIV, or anti-cancer effects [1,2]. Betulin is non-toxic and may be related to the class of low-toxic substances [3]. However, betulin is only slightly soluble in water (as it was reported in [4], the solubility of betulin is 0.08 μg/mL), and this substantially limits its bioavailability. Several strategies have been applied to improve the solubility and/or dissolution rate of betulin and its derivatives, such as preparation of solid dispersions with water-soluble polymers [5], cyclodextrin derivatives [6], as well as entrapment in nanosystems [7,8]. For example, in [7,8], nanosystems with betulin were obtained, which exhibited high bioavailability and had optimal physicochemical properties for inhalation administration. However, preparation of amorphous systems or particle size reduction cause systems instability, so different approaches are applied for their stabilization. An alternative approach available for the enhancement of drug solubility, dissolution rate and bioavailability involves the application of crystalline drug forms, such as polymorphs, solvates, hydrates, and cocrystals, which have suitable physical and physicochemical characteristics. For example, the main advantages of cocrystals over the amorphous forms

of the drugs for solubility improvement are that they exist in a stable crystalline form and do not require other excipients and additives in formulations [9].

Since the physicochemical properties of drugs, including solubility and dissolution kinetics, are largely determined by their crystal structure and particle size distribution, the polymorphic forms in which they occur are of specific importance [10,11]. Thus, for the preparation of betulin-based drug forms, the polymorphism of betulin is one of the crucial aspects.

Recently [12], it was shown that betulin, crystallized from solution or prepared by decomposition of ethanol solvate at room temperature, is a hemihydrate of botulin. By means of in situ X-ray powder diffraction, it was found that when betulin hemihydrate (betulin III) was heated, a number of structural transformations proceeded in addition to water removal. Until 110 °C, the unit cell parameters smoothly changed due to dehydration. In the temperature range from 110 °C to 120 °C, there was a transition to an intermediate crystalline form (betulin II), which transformed into the orthorhombic form of pure betulin (betulin I) [13] before melting. Betulin I could also be prepared by decomposition of betulin ethanol solvate at 200 °C [12,13].

In [14], three crystalline forms of betulin were also found. It should be noted that the X-ray powder diffraction patterns of these forms were very similar to those obtained in [12].

The pseudopolymorphic modifications of betulin are also known: in addition to betulin hemihydrate, there are also its solvates [15–18]. In our previous works [19–23], some new crystalline forms of betulin in the mixtures of betulin with dicarboxylic acids (adipic, suberic, or terephthalic) were obtained with the help of the liquid-assisted grinding (LAG) method [24]. These forms were attributed to the cocrystals of betulin. They revealed improved solubility in comparison with initial betulin. It was shown that betulin cocrystals with adipic and suberic acids could also be prepared by heating the mixtures of the components up to the melting points of the acids [23,24].

Thus, under heating, betulin can transform into another polymorphic form and, moreover, it can interact with dicarboxylic acids forming cocrystals. These transformations should be accompanied by changes in the crystal morphology. On the other hand, the changes in crystal morphology could indicate a change in the crystalline form of betulin. In turn, this leads to a change in the physicochemical properties of betulin, including pharmaceutically relevant ones, such as dissolution rate and solubility.

In this paper, the morphological changes that occur during heating of betulin alone and in the presence of dicarboxylic acids were investigated. Adipic and suberic acids were chosen for this study. Along with X-ray powder diffraction measurements and electron microscopic investigations of the morphology of the obtained powders, their densities were also measured. Gas pycnometers allow measuring the densities of solids with the reproducibility of 0.01%, which in most cases makes it possible to reliably distinguish between different crystalline forms of the same substance. The rates of dissolution of the samples obtained by betulin heating at different temperatures were compared using a water–ethanol solution as the dissolution medium.

## 2. Materials and Methods

Betulin (Figure 1a) was isolated from birch bark according to the developed method [25] and purified by crystallization from ethanol (purity is 97.7%). Adipic acid (Figure 1b) (Riedel-de Haen AG, Bucsh, Switzerland) and suberic acid (Figure 1c) (Reachim, JSC, Moscow, Russia) were of analytical grade and were used as received.

**Figure 1.** Molecular structures of betulin (**a**), adipic acid (**b**), and suberic acid (**c**).

Cocrystals of betulin with adipic and suberic acids were prepared with the help of the LAG method as described previously [20]. Physical mixtures of betulin with the acids (at a molar ratio of 1:1 or 1:0.13) were prepared by mixing the powders of the components thoroughly with a needle. Betulin or its mixtures with the acids were previously heated in the heating chamber up to definite temperature, and then kept at this temperature for 10 min. The error of temperature measurement was 5 °C. After that, the morphology was studied under electron microscope. Scanning electron microscopy (SEM) studies were performed with a TM-1000 Hitachi (Hitachi Scientific Ltd., Tokyo, Japan) microscope.

X-ray powder diffraction (XRPD) measurements were carried out on a D8 Advance diffractometer (Bruker, Germany) with a Lynx-Eye one-dimensional detector, CuK$\alpha$-radiation, 2$\theta$ = 5–70°. High-temperature measurements were carried out using a high-temperature chamber HTK 1200N (Anton Paar, Graz, Austria). A powder sample was placed in a corundum carrier and heated in air at a heating rate of 0.2 °C/s. The samples were kept at each selected temperature for 10 min.

Calorimetric measurements (DSC) were performed using a DSC 200 F3 MAIA (Netzsch, Selb, Germany) heat flux calorimeter. A solid sample, about 5 mg, was placed in 40 µL closed aluminum caps and scanned from 25 to 250 °C under argon atmosphere at a heating rate of 6 °C/min.

Densities of the powder samples were measured using an UltraPyc 1200e (Quantachrome Instruments, Seattle, DC, USA) helium gas pycnometer. A cell 10 cm$^3$ in volume was used. The accuracy and repeatability of metering were better than 0.03% and 0.015%, respectively.

The dissolution behavior of betulin samples was investigated using a 705 DS Dissolution Apparatus (Varian, INC, Cary, NC, USA). Dissolution experiments were carried out at 30 °C using a paddle rotating at 75 rpm. An amount of 10 mg of the sample was put into 50% (by volume) solution of ethanol in water, which was used as a dissolution medium. The 1 mL probes were sampled from the solution at 3, 8, 15, 25, 45, 60 and 90 min adding the appropriate volume of the solvent to the solution after each probe sampling. The sample solution was filtered through the blue filter paper, and the filtrate was directly injected into high-performance liquid chromatography (HPLC) system and assayed for betulin concentration. HPLC analysis was performed using a Milichrom A-02 chromatograph (EcoNova LLC, Novosibirsk, Russia) with a ProntoSIL 120-5-C18 AQ analytical column (2.0 mm × 75 mm i.d., 5 µm particle size). The mobile phase consisted of water (A) and acetonitrile (B). The gradient was as follows: 0–3 min, 80–100% (solvent B), flow rate: 0.2 mL/min; 4–12.5 min, 100–100% (solvent B), flow rate: 0.2 mL/min. The column temperature was maintained at 35 °C, and the effluent was monitored at 200, 210, and 220 nm.

## 3. Results and Discussion

### 3.1. Morphology Changes during the Polymorphic Transformations of Betulin

Figure 2 shows the DSC thermogram of betulin used in this work. It is in good agreement with the DSC curve of betulin form III given in the previous paper [12]. The thermal effect at low (up to 120 °C) temperatures can be explained by the gradual loss of

water molecules without abrupt structural changes. The sharp endothermic peak with $T_{onset}$ = 232 °C corresponds to the melting of orthorhombic betulin [12,13]. Some thermal effects before melting can be explained by solid-phase transformations as it was observed in the X-ray powder thermodiffraction experiments [12].

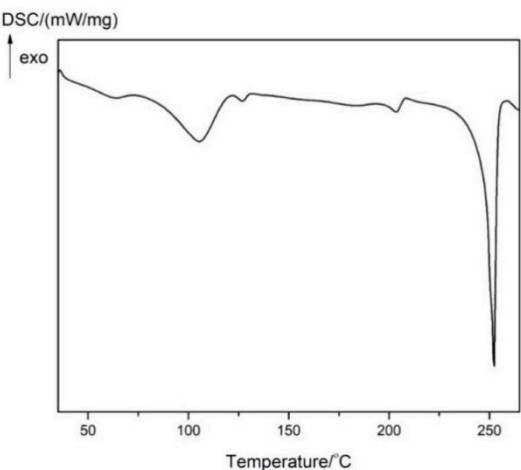

**Figure 2.** DSC thermogram of betulin.

The morphology of betulin crystals in the temperature range of 110–180 °C was followed in order to elucidate the morphological changes of betulin during polymorphic transformations. The starting betulin was composed of large prismatic crystals. Figure 3 shows that under heating up to 120 °C, the prismatic crystals were converted into small needle-like crystals, whereas near 180 °C long whiskers were grown.

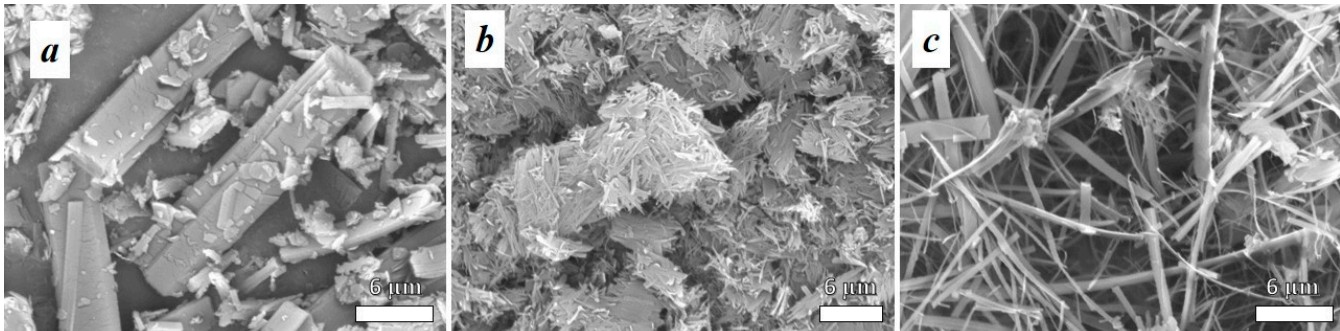

**Figure 3.** SEM images of the starting betulin (**a**) and after heating up to 120 °C (**b**) and 180 °C (**c**). Scale line is 6 μm.

In Figure 4, the XRPD patterns of betulin heated from room temperature to 140 °C are shown. At 100 °C, the reflexes slightly changed, probably due to water removal, and under subsequent heating, the formation of betulin II polymorph was observed. The diffraction pattern taken after cooling was similar to that of betulin II [12].

Table 1 shows the results of measuring the densities of the samples after heating up to 110 and 130 °C. The density of the crystals decreased in both cases. Furthermore, one can see that the difference in crystal density of different forms is 20–30 times higher than the reproducibility of determining the density of each phase. This can be considered as additional evidence that, when heated to 130 °C, betulin undergoes at least two structural transformations, namely crystal transformation due to the removal of water, and the second one which is associated with the transition to polymorphic form II.

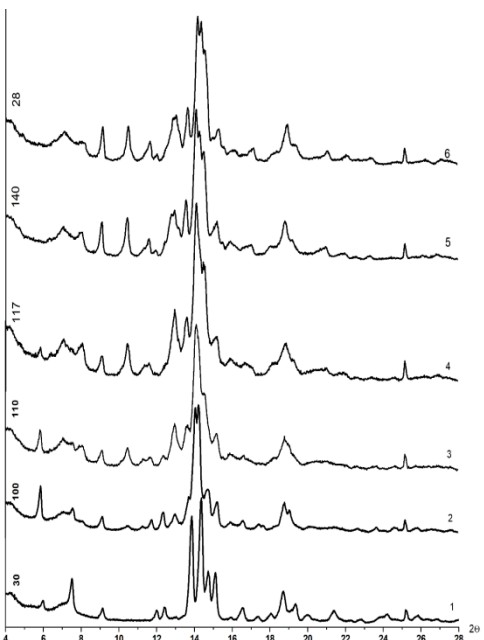

**Figure 4.** XRPD patterns of betulin taken in situ after heating up to 30 °C (1), 100 °C (2), 110 °C (3), 117 °C (4), 140 °C (5), and cooling to 28 °C (6).

**Table 1.** Density of betulin polymorphs.

| Polymorphic Form | Heating Temperature, °C/ Crystal Shape | Density Measured at 20 °C, g/cm$^3$ | Variation Coefficient, % |
|---|---|---|---|
| III (hemihydrate of betulin) | RT/prismatic crystals | 1.1147 ± 0.0016 | 0.14 |
| Anhydrous betulin | 110/prismatic crystals | 1.0704 ± 0.0015 | 0.14 |
| II | 130/needle-like crystals | 1.0303 ± 0.0042 | 0.41 |

It is known that solid-state reactions are often accompanied by fracture as a result of product nuclei formation and relaxation of stresses at the interface [26]. Thus, the formation of needle-like crystals can be associated with a fracture of initial crystals induced by polymorphic transformation. At the same time, after dehydration, the shape of the crystals did not change significantly despite the fact that the density decreased. This is consistent with the fact that, as was shown in the previous paper [12], until 110 °C betulin unit cell parameters smoothly changed. It can be assumed that water molecules, due to their location in the crystal in the form of stacks, could leave the crystal without significant structural disturbance.

*3.2. Morphology Changes during Betulin Heating in the Presence of Dicarboxylic Acids*

3.2.1. Morphology Changes in the Mixtures of Betulin with Adipic Acid

It was found that in the case of a physical mixture of betulin with adipic acid heated up to 110–120 °C, the morphology of betulin crystals differed from that of betulin heated alone. Indeed, Figure 5 shows that the plates were observed instead of the needles after betulin heating in the mixture with adipic acid.

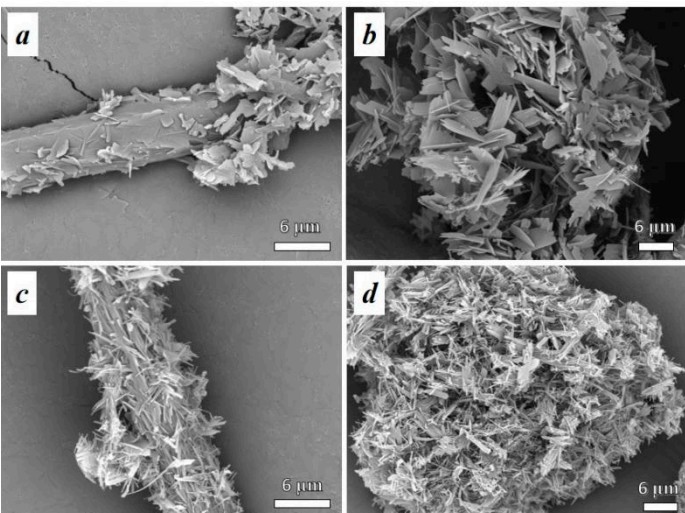

**Figure 5.** SEM images of betulin after heating up to 120 °C in the mixture with adipic acid (**a**,**b**) in comparison with betulin alone (**c**,**d**). Scale line is 10 µm.

The XRPD patterns of 1:1 betulin–adipic acid physical mixture taken at different temperatures are shown in Figure 6. A new crystalline product is observed to be formed under heating of 1:1 betulin–adipic acid physical mixture. The reflexes of initial reagents disappeared, and new reflexes were observed. Nevertheless, the acid peaks do not disappear completely, suggesting that not the whole amount of the acid interacted with betulin to form a new crystal structure.

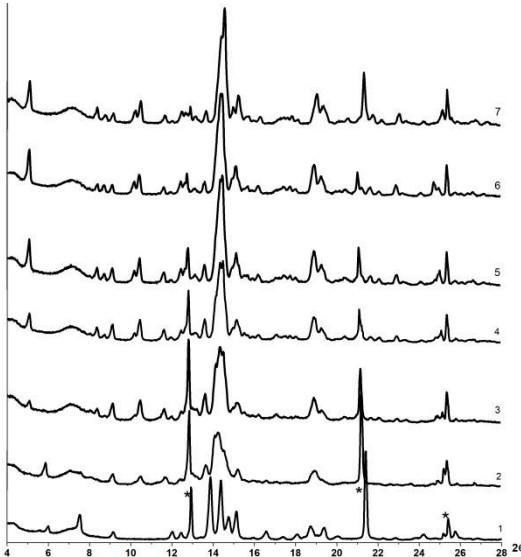

**Figure 6.** XRPD patterns of the 1:1 betulin–adipic acid mixture taken in situ after heating up to 30 °C (1), 100 °C (2), 110 °C (3), 117 °C (4), 125 °C (5), 140 °C (6) and cooling to 28 °C (7). (The reflexes of adipic acid in the starting mixture are denoted by asterisks.).

The XRPD patterns of the heated betulin–adipic acid physical mixture were similar to those of the mixtures ball milled in the presence of a liquid with the formation of cocrystals [23] (Figure 7).

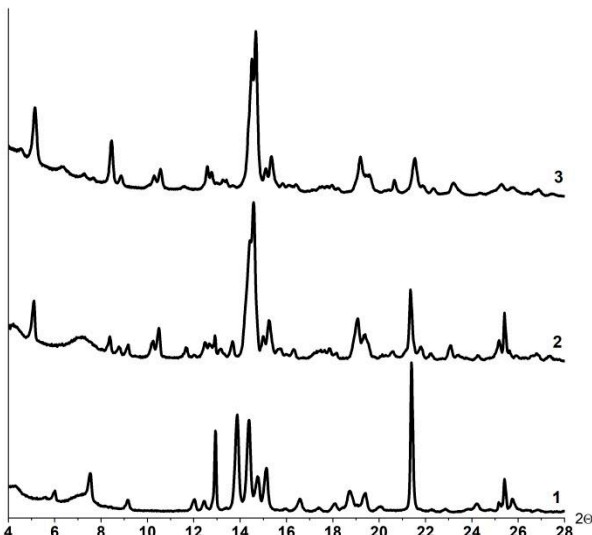

**Figure 7.** XRPD patterns of 1:1 betulin–adipic acid physical mixture before (1) and after heating up to 140 °C (2); 1:1 betulin–adipic acid mixture after processing by the LAG method (3).

At this stage of the study, it is not possible to identify the structures of the resulting betulin compounds with adipic and suberic acids. Therefore, hereinafter, we provide only XRPD patterns of the resulting mixtures. The possibility of obtaining new compounds may be indicated by morphological data, the densities of the obtained compounds, as well as by a change in solubility.

It can be assumed that the interaction under heating occurs through the gas phase. To demonstrate the possibility of adipic acid sublimation at a temperature of 100–110 °C, a special experiment was performed using a tube vial with a hot section and a cold one. When adipic acid was placed in the vial in its hot section with a temperature of 100–110 °C, crystallization of the acid was observed in the cold section (15 °C). Furthermore, we noticed that when the 1:1 betulin–adipic acid physical mixture was heated up to 140 °C and then cooled, the excess of the acid crystallized from gas phase, which was easily observed in the SEM image (Figure S1). After washing this sample with water, the acid peaks disappeared from the X-ray diffraction pattern, while the peaks related to betulin cocrystal were still present (Figure S2).

The polymorphic transition of betulin within this temperature range probably facilitates the interaction of betulin with the acid according to the Hedvall effect [27], meaning a solid-state chemical reaction that proceeds faster in the region of a phase transition. Nevertheless, the interaction was not complete because, in contrast to the grinding method, there was no mixing during heating, and therefore not the entire surface of the crystals was available for the gas phase.

It could be assumed that the acid affected the shape of betulin crystals during their polymorphic transition. In this case, a small amount of acid should be sufficient for morphological changes to occur. However, after heating a mixture of betulin with adipic acid at a ratio of 1:0.13 (by mole) up to 140 °C, a mixture of phases was observed in the SEM micrograph, namely betulin in the form of the needles and in the form of the plates, i.e., a part of betulin passed into polymorph II, and a part of it interacted with adipic acid forming cocrystal (Figure 8). In Figure 9, the XRPD pattern of the 1:0.13 betulin–adipic acid mixture after heating up to 140 °C (curve 2) actually shows the peaks of both betulin phases.

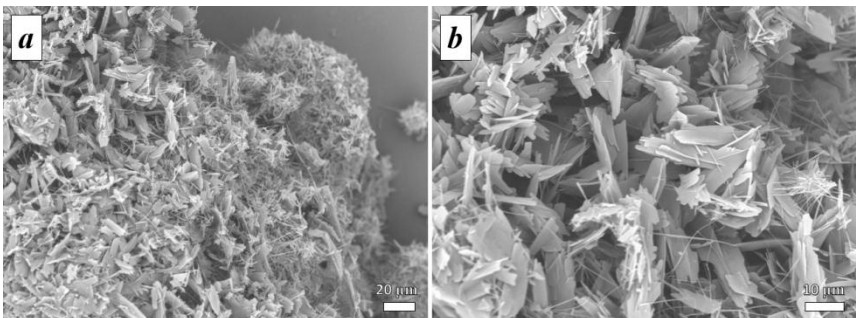

**Figure 8.** SEM images of 1:0.13 betulin–adipic acid physical mixture before (**a**) and after heating up to 140 °C (**b**). Scale line is 20 μm (**a**) and 10 μm (**b**).

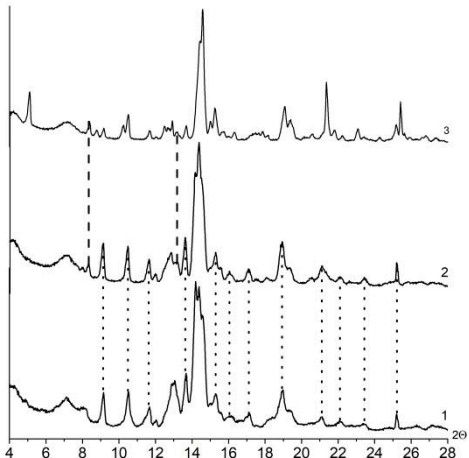

**Figure 9.** In situ XRPD patterns of betulin (1), 1:0.13 (2) and 1:1 (3) betulin–adipic acid physical mixtures after heating up to 140 °C. In pattern 2, dotted lines mark reflections of two phases.

### 3.2.2. Morphology Changes in the Mixtures of Betulin with Suberic Acid

Figure 10 shows SEM images of a 1:1 betulin–suberic acid physical mixture after heating up to 120 °C in comparison with betulin alone. One can see that the needle form of the crystals was practically identical for betulin and for its mixture with suberic acid.

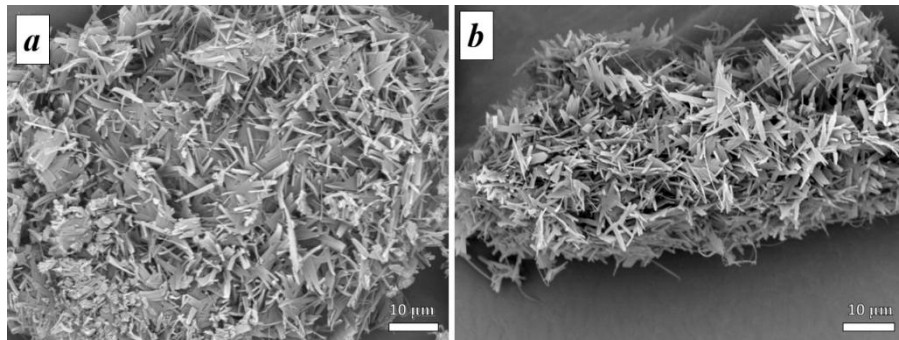

**Figure 10.** SEM images of betulin (**a**) and 1:1 betulin–suberic acid physical mixture (**b**) after heating up to 120 °C. Scale line is 10 μm.

In the X-ray diffraction patterns of the physical mixture of betulin with suberic acid taken under heating (Figure 11), the structural changes were observed in comparison with the initial reagents. At the same time, XRPD patterns of the mixtures of betulin with suberic acid processed by the LAG method and after heating (curves 2 and 3 in Figure 12) are different. This suggests that in the case of suberic acid, the interaction through the gas phase does not occur. This may be due either to the fact that suberic acid does not sublime

to a sufficient extent at these temperatures or to the fact that suberic acid molecules, which are larger than the molecules of adipic acid, experience steric hindrance for attack from the gas phase. As a result, betulin passes into its polymorphic modification, which can be seen both in the shape of the crystals (Figure 10) and in XRPD patterns (curves 1 and 2 in Figure 11). The changes in the reflexes were also caused by the phase transitions in suberic acid [28], as is seen in Figure S3.

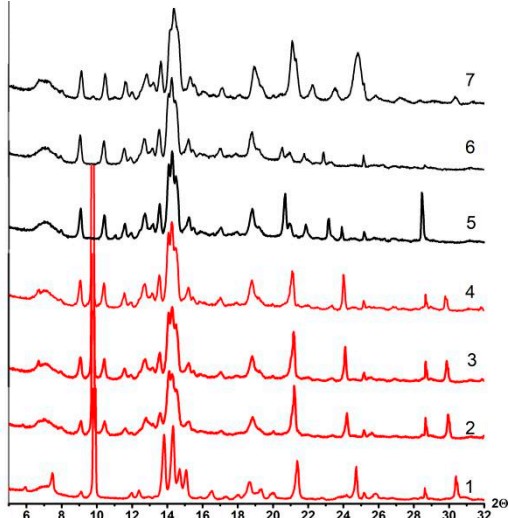

**Figure 11.** XRPD patterns of 1:1 betulin–suberic acid physical mixture taken in situ after heating up to 29 °C (1), 100 °C (2), 110 °C (3), 117 °C (4), 125 °C (5), 130 °C (6) and cooling to 27 °C (7). The 1–4 patterns are marked in red to show phase transformations in suberic acid.

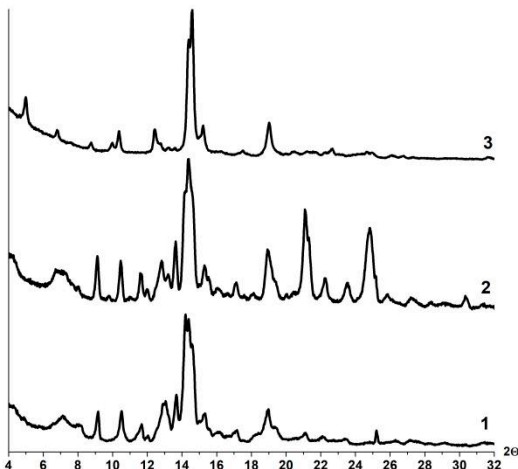

**Figure 12.** XRPD patterns of the samples heated up to 130 °C and then cooled to 28 °C. 1–betulin, 2–1:1 betulin–suberic acid physical mixture, 3–1:1 betulin–suberic acid mixture after processing by the LAG method.

### 3.3. Dissolution of Betulin Polymorphs Obtained by Heating of the Starting Substance

Taking into account the fact that the large surface area of the samples shaped as needles or whiskers, obtained under betulin heating, could result in an increase in the rate of betulin dissolution, we studied the kinetics of dissolution of the samples obtained at different temperatures. The formation of whiskers of betulin at 180 °C was especially interesting, suggesting the possibility to increase the rate of betulin dissolution.

To study the dissolution rate of betulin polymorphs, water–ethanol binary solvents with different component ratios were taken, and the solvent with 50% (by volume) content was chosen, since in this case, the difference in the dissolution rate of various polymorphs

was acceptable for analysis. Typical chromatograms of betulin in water–ethanol solution are shown in Figure 13.

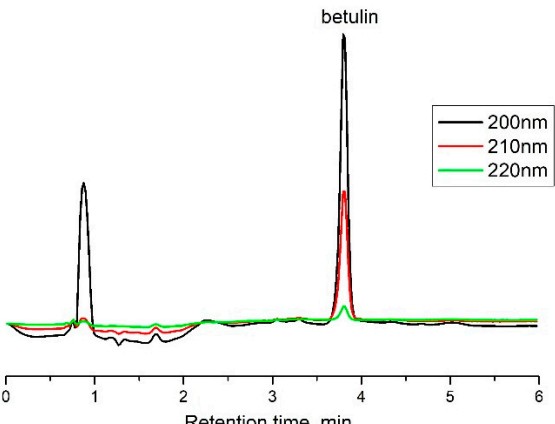

**Figure 13.** Typical HPLC chromatograms of betulin in water–ethanol solution.

Figure 14 shows the dissolution curves of betulin polymorphs, presented as the percentage of dissolved betulin versus dissolution time. One can see that dissolution profiles are different as evidenced by the error bars. As particle dimensions and therefore the surface area of the samples are different, a direct comparison of intrinsic dissolution rate and solubility of the polymorphs is hindered. Despite this, it is possible to highlight a faster dissolution rate during the first stage of the study and a successive decrease in the dissolution rate for all phases.

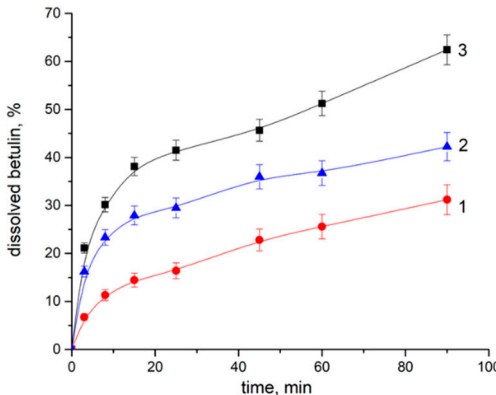

**Figure 14.** Dissolution curves of betulin samples prepared by heating the starting substance at 130 °C (1), 200 °C (2), and hemihydrate of betulin (3). Experiments were performed in triplicate.

In the majority of dissolution phenomena, the solvation step is almost instantaneous, and the diffusion process constitutes the rate-limiting step [29]. Nevertheless, in our case, taking into account the fact that the substances under study have low dissolution rates and the process is carried out under intense stirring, we could assume that, at least at the earliest stage of dissolution (during the first 10 min), the process takes place at the interface, i.e., in the kinetic mode. Thus, using the measured surface areas, the dissolution rate constants have been estimated to be form I–$1.98 \times 10^{-3}$ mg·min$^{-1}$·cm$^{-2}$; form II–$1.01 \times 10^{-3}$ mg·min$^{-1}$·cm$^{-2}$; form III–$5.16 \times 10^{-4}$ mg·min$^{-1}$·cm$^{-2}$.

It can be seen that whiskers formed at 200 °C exhibit an increased dissolution rate compared to polymorph II formed at 130 °C. However, both factors, such as the higher dissolution rate constant and the higher surface area, can lead to an increase in the dissolution rate. Figure 14 shows that the starting betulin form III, which is a hemihydrate of betulin, exhibits the highest dissolution rate. It should be noted that, as reported for different APIs [30,31], the anhydrous phases could be solubilized more rapidly than their

respective hydrate forms. Nevertheless, in general, the presence of water molecules affects the level of intermolecular interactions and the degree of crystalline disorder, thus affecting the solubility and dissolution rate of hydrated APIs [32]. In particular, the substitution of homomolecular contacts in a crystal for heteromolecular ones (for example, by preparing cocrystals or hydrate forms) may be one of the ways to facilitate dissolution [33]. Thus, we could suggest that in the hemihydrate of betulin, heteromolecular interactions could be weaker than homomolecular ones, facilitating the dissolution. Moreover, the dissolution rate of hydrated betulin was higher, probably due to the higher specific surface area.

## 4. Conclusions

In this paper, the changes in betulin morphology during heating at selected temperatures corresponding to its polymorphic transformations were investigated. The results showed that the prismatic crystals of starting betulin form III converted into needles at 120 °C as a result of water removal and III→II polymorphic transition. At further heating up to 180 °C, the whiskers of betulin form I were grown.

The experiments on betulin heating in the presence of dicarboxylic acids, adipic or suberic reveal that the morphological changes can serve as a test for cocrystal formation. According to morphological changes, the formation of cocrystals of betulin with adipic acid under heating was identified prior to acid melting. The interaction of adipic acid vapor with the surface of betulin was supposed. In contrast to this, the morphological changes in betulin–suberic acid mixture during heating provided only evidence of the polymorphic transformations of the components.

In this work, the formation of betulin whiskers was discovered for the first time. The formation of the whiskers of betulin at heating up to 180 °C is especially interesting because one could expect new properties in the nanodispersed state of the drug. Indeed, the rate of dissolution of this polymorph was higher than that of polymorph II. The detailed mechanism of the formation of betulin whiskers and their properties will be studied in the future.

It is known that the solubilities of various polymorphs and solvatomorphs are different, and these differences may lead to differences in the dissolution rate, which in turn could lead to differences in bioavailability. The dissolution rate depends also on the particle size and surface area of the solid. The study of dissolution of betulin polymorphs obtained by heating showed that despite the fact that polymorphic modifications with large surface area were formed when betulin was heated, the starting betulin polymorph III revealed the highest dissolution rate in comparison with betulin polymorphs obtained under heating. This may be due to the fact that heteromolecular interactions in betulin hemihydrate are weaker than the homomolecular ones in the anhydrous betulin polymorphs, as well as due to the higher specific surface area.

**Supplementary Materials:** The following supporting information can be downloaded at: https://www.mdpi.com/article/10.3390/powders2020026/s1, Figure S1: SEM image of 1:1 betulin–adipic acid physical mixture after heating up to 140 °C. Acid crystals are shown with an arrow. Scale line is 10 μm; Figure S2. XRPD patterns of adipic acid (1), 1:1 betulin–adipic acid physical mixture heated up to 140 °C (2); the same sample after washing with water (3); Figure S3. XRPD patterns of suberic acid taken in situ after heating up to 29 °C (1), 100 °C (2), 110 °C (3), 117 °C (4), 125 °C (5), 130 °C (6) and cooling to 27 °C (7).

**Author Contributions:** Conceptualization, S.A.M. and A.A.P.; methodology, S.A.M.; software, S.A.M.; validation, S.A.M. and A.A.P.; formal analysis, S.A.M.; investigation, S.A.M.; resources, S.A.K.; data curation, T.P.S.; writing—original draft preparation, T.P.S.; writing—review and editing, T.P.S.; visualization, S.A.M.; supervision, T.P.S.; project administration, T.P.S.; funding acquisition, T.P.S. All authors have read and agreed to the published version of the manuscript.

**Funding:** This research was funded by the budget projects: No. 121032500064-8 for the Institute of Solid State Chemistry and Mechanochemistry SB RAS and No. 121031500180-8 for the Institute of Chemistry and Chemical Technology SB RAS.

**Institutional Review Board Statement:** Not applicable.

**Informed Consent Statement:** Not applicable.

**Data Availability Statement:** The data presented in this study are available on request from the corresponding author. The data are not publicly available because they are part of an ongoing study.

**Conflicts of Interest:** The authors declare no conflict of interest.

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
