# Peer review of "Morphological Changes in Betulin Particles as a Result of Polymorphic Transformations, and Formation of Co-Crystals under Heating"

_2674-0516, doi:10.3390/powders2020026_

Round 1

Reviewer 1 Report

In the present work, the authors try to discuss the relationship between the morphological changes during milling and subsequent heating of botulin crystals and their chemical and crystallographical properties. The manuscript is carefully prepared and reasonably understandable. Therefore, the manuscript might be recommended for publication.

On the other hand, there are several open questions, which should be answered and the manuscript should be revised accordingly. All the discussions made are qualitative. The most important explanation among the cross relationship among the morphology and crystallography due to milling and subsequent heating is by far insufficient. Most of the results are illustrated in terms of XRPD and SEM. The associated interpretation is not sufficiently explained. Explicit points are listed below for the authors’ sake for revision.

(1)  Interpretation and assignments of XRPD: The authors display 6 Figures, 4,6,7,9,11, and 12. There are, however, no assignments and no discussion of lattice constants or lattice disturbances. Therefore, the discussion is just qualitative. Minimum discussion in these respects is indispensable.

(2)  Morphological changes: They are associated with the change in the non-isotropic intercrystalline forces. For instance, development of needle-like morphology from Fig. 3b to 3c by increasing the temperature, should be associated with the development of unidirectional interlattice forces. Most probably, these changes are associated with the change in the state of cocrystals.

(3)  Change in the rate of dissolution rate: This is one of the most important items in physical pharmaceutic aspects. In Fig. 13, the authors display that the rate of dissolution increased by heating temperature and approaches to that of hemihydrate. The authors are aware that the kinetics reflect the specific surface area. There are many quantitative methods to correct, if in part, the effects. A simplest possible method might be to analyze the kinetics, obtain the rate constant, and divide by the specific surface area.

The reviewer expects the authors to make a major revision to make the manuscript acceptable by improving the discussion more explicit and quantitative.

Reviewer 2 Report

The article investigates the changes in betulin crystals during heating and how the morphological changes can serve as a test for cocrystal formation, with the formation of a cocrystal of betulin with adipic acid being identified, but despite the formation of polymorphic forms with a high surface area during heating, dissolution studies showed that the starting betulin polymorph III had the highest dissolution rate. The manuscript is well written, but the authors must address some concerns before it is considered for publication.

1) Line 31, could you list the detailed solubility of betulin in the introduction part?

2) Could you please add the value of the scale lines in all SEM images? It would be easier for readers to understand the size of your cocrystals.

3) Please ensure that all XRD patterns are presented uniformly in the same template, as some patterns lack x-axis titles or have improperly positioned x-axis titles.

4) Have you tried to grow single crystals of your products? Is it possible to get single crystals structures of them?

5) Could you also provide some characterizations of your cocrystals such as NMR, TGA, DSC, etc?

Reviewer 3 Report

After careful evaluation of the manuscript by Svetlana A. Myz et al I have next recommendations and comments:

1. In Introduction: please provide information about the toxicity of betulin.

2. The bioavailability of betulin was increase by preparation of nanosystems containing betulin for inhalation  and by preparation of Nanosystem-Entrapped Betulin for Endotracheal Administration. Please update the introduction.

3. In Sect. 2: please indicate the purity of betulin.

4. Which reference compound was used for HPLC analysis?

5. How many polymorphous form of betulin has been described in the literature?

6. In Fig. 2 please clarify if this is DSC thermogram of extracted betulin?

7. In legend to Fig 11: please explain why some XRPD are marked in red?

8. In Sect.3.3 please provide typical HPLC chromatograms.

9. Discussion is weak. Authors must to compare own results with previously published. For example please compare release profiles of betulin

10. Please explain the reasonobility for cocrystals of betulin with adipic  and suberic acids?with other formulations (e.g nanoformulations).

Round 2

Reviewer 1 Report

The revision was done in an appropriate manner.

Reviewer 3 Report

The manuscript could be accepted in present revised form.